# Marine Bacteria from Rocas Atoll as a Rich Source of Pharmacologically Active Compounds

**DOI:** 10.3390/md17120671

**Published:** 2019-11-28

**Authors:** Karen Y. Velasco-Alzate, Anelize Bauermeister, Marcelo M. P. Tangerina, Tito M. C. Lotufo, Marcelo J. P. Ferreira, Paula C. Jimenez, Gabriel Padilla, Norberto P. Lopes, Letícia V. Costa-Lotufo

**Affiliations:** 1Departamento de Farmacologia, Instituto de Ciências Biomédicas, Universidade de São Paulo, 05508-900 São Paulo/SP, Brazil; karenvelasco20@gmail.com (K.Y.V.-A.); ane.meister@usp.br (A.B.); 2NPPNS, Departamento de Física e Química, Faculdade de Ciências Farmacêuticas de Ribeirão Preto, Universidade de São Paulo, 14040-903 Ribeirão Preto/SP, Brazil; npelopes@fcfrp.usp.br; 3Departamento do Botânica, Instituto de Biociências, Universidade de São Paulo, 05508-090 São Paulo/SP, Brazilmarcelopena@ib.usp.br (M.J.P.F.); 4Departamento de Oceanografia Biológica, Instituto Oceanográfico, Universidade de São Paulo, 05508-120 São Paulo/SP, Brazil; tmlotufo@usp.br; 5Departamento de Ciências do Mar, Universidade Federal de São Paulo, 11015-020 Santos/SP, Brazil; paulacjimenez@gmail.com; 6Departamento de Microbiologia, Instituto de Ciências Biomédicas, Universidade de São Paulo, 05508-900 São Paulo/SP, Brazil; gpadilla@icb.usp.br

**Keywords:** secondary metabolites, microbial diversity, metabolomics, molecular network, marine bacteria

## Abstract

Rocas Atoll is a unique environment in the equatorial Atlantic Ocean, hosting a large number of endemic species, however, studies on the chemical diversity emerging from this biota are rather scarce. Therefore, the present work aims to assess the metabolomic diversity and pharmacological potential of the microbiota from Rocas Atoll. A total of 76 bacteria were isolated and cultured in liquid culture media to obtain crude extracts. About one third (34%) of these extracts were recognized as cytotoxic against human colon adenocarcinoma HCT-116 cell line. 16S rRNA gene sequencing analyses revealed that the bacteria producing cytotoxic extracts were mainly from the Actinobacteria phylum, including *Streptomyces*, *Salinispora*, *Nocardiopsis*, and *Brevibacterium* genera, and in a smaller proportion from Firmicutes phylum (*Bacillus*). The search in the spectral library in GNPS (Global Natural Products Social Molecular Networking) unveiled a high chemodiversity being produced by these bacteria, including rifamycins, antimycins, desferrioxamines, ferrioxamines, surfactins, surugamides, staurosporines, and saliniketals, along with several unidentified compounds. Using an original approach, molecular networking successfully highlighted groups of compounds responsible for the cytotoxicity of crude extracts. Application of DEREPLICATOR+ (GNPS) allowed the annotation of macrolide novonestimycin derivatives as the cytotoxic compounds existing in the extracts produced by *Streptomyces* BRB-298 and BRB-302. Overall, these results highlighted the pharmacological potential of bacteria from this singular atoll.

## 1. Introduction

Marine environments are a rich source of genetic diversity, further translated through the biosynthesis of unique and complex chemical structures with biotechnological and pharmacological interest [1,2]. During the last decade, around 1000 new natural products have been described each year and, only in 2016, for instance, 1277 new compounds were reported in 432 papers [1]. This chemical diversity arises from a great variety of organisms, such as microorganisms, algae, tunicates, sponges, mollusks, bryozoans, and cnidarians, among many others, making marine environments particularly interesting to be deeply investigated in order to discover new compounds with promising therapeutic properties [1,2].

Brazil has the second largest continuous coastline in the world, with circa 8500 km of extension and over 3.5 million km^2^ of economic exclusive zone [3]. It also includes five sets of tropical oceanic islands—Fernando de Noronha Archipelago (FNA), Trindade Island and Martim Vaz Archipelago, Rocas Atoll, and Saint Peter and Saint Paul Archipelago (SPSPA)—and is therefore favorable to the development of a rich and diverse biota. SPSPA, Rocas Atoll, and FNA are located in the equatorial zone of the Atlantic, and although they form a set of islands with a great number of overlapping biotic components, each hosts peculiar faunas and cases of endemism [4,5].

Atolls are oceanic islands with volcanic and biogenic formation. Their structure is usually ring-like, forming a lagoon inside with little water exchange, resulting in the development of biological reefs. Such conditions make these environments favorable to host a diverse biota, and consequently, a rich source of marine natural products [4,5,6]. This kind of natural formation is rather common in the Pacific and Indian Oceans, however, in the Atlantic Ocean, there are merely Rocas Atoll, in northeastern Brazil, and a few more in the Caribbean Sea. Rocas Atoll was the first acknowledged marine biological reserve in Brazil, granted in 1979, being a highly preserved marine environment [7]. Located 260 km off the coast, Rocas Atoll is a unique environment, hosting many endemic species [4,5]. For instance, Netto et al. [8] described 95 taxa from the meiofauna and 79 macrofaunal taxa from the sediments of the tidal pools alone. A previous study from our research group described the occurrence of 12 ascidian species from Rocas Atoll, out of which five were recognized as new species [4].

Ascidians and sponges are filter-feeding invertebrates found in most marine environments, including Rocas Atoll. These organisms have yielded a wide variety of chemical structures, including antitumor compounds, with more than 6000 natural products described so far [9,10,11]. However, the production of most of these metabolites was later attributed to the invertebrate-associated bacteria living in a syntrophic relationship. These symbiotic bacteria produce different chemical structures, generally in response to interactions (antagonistic or beneficial) with their host [12,13]. These invertebrate-associated microbial communities are considered complex ecosystems, called holobionts, which comprise a high microbial diversity, composed predominantly of proteobacteria, but also of several other groups of bacteria and archaea [14,15,16,17,18].

Therefore, the goal of the present work is to assess culturable microorganisms from Rocas Atoll and analyze the cytotoxicity and the metabolomic profile of organic extracts produced by these bacteria, considering the enormous pharmacological potential described for marine bacteria. To accomplish this goal, the following proposal was designed: (1) to recover culturable bacteria from ascidians, sponges, and sediments from Rocas Atoll; (2) to test the cytotoxicity of the bacterial crude extracts in colon carcinoma cells (HCT-116); and (3) to investigate the chemical content of chemical extracts, in order to characterize the metabolic diversity and to define possible relations with cytotoxicity properties.

## 2. Results and Discussion

### 2.1. Bacteria Recovered from Rocas Atoll: Cytotoxicity and Identification

Considering the great pharmacological and biotechnological potential of the metabolites produced by marine bacteria [17,19], in this work, we pursued the prospection of the culturable bacteria from Rocas Atoll, focusing largely on actinomycete-like strains and their resulting metabolomic diversity. Nine ascidians and four sponges, along with seven sediment samples, were collected, aiming at broadening the assortment of the assessed microbiota. Appendix A shows the recovered bacteria from ascidians (39%), sponges (36%), and sediments (25%). The higher percentage of culturable bacteria was found in the invertebrates, especially from *Trididemnum maragoii*, an endemic ascidian from Rocas Atoll, and from the sponge *Chondrilla* cf. *nucula*, thus highlighting these holobionts as rich microbial microenvironments. Appendix A describes the bacterial strains obtained, including their origin (ascidian, sponge, or sediment), collection site, and taxonomic identification. 

The cytotoxic activity evaluation of the ethyl acetate extract obtained from each strain showed that 26 crude extracts were considered cytotoxic at 50 µg/mL (Figure 1a), while nine were also active at 5 µg/mL (Figure 1b). These numbers are similar to those obtained from other studies performed with marine bacteria recovered from sediments collected at Atlantic islands [20,21].

In a previous work that investigated culturable actinomycetes isolated from sediments collected at the Saint Peter and Saint Paul Archipelago, 268 strains were isolated from 21 sediment samples, while 94 strains were selected for cytotoxicity analysis of their crude extract against HCT-116 cell lines [20]. Among them, 26 strains produced cytotoxic extracts, and the chemical analysis by liquid chromatography- mass spectrometry (HPLC-MS/MS) suggested the production of known cytotoxic compounds, such as staurosporines and piericidins [20]. 

Herein, only strains that produced cytotoxic extracts were selected for taxonomic identification by 16S rRNA gene sequencing. The bacteria belonged to two phyla: Actinobacteria and Firmicutes (Endobacteria) (Figure 2). Among the identified bacteria, Actinobacteria comprised nearly 87%, being the dominant phylum among the producers of cytotoxic extracts. Within Actinobacteria, Micromonosporales and Streptomycetales were the most abundant taxa (40.0% and 35.0%, respectively, across all classified bacteria), followed by Streptosporangiales (20.0%) and Actinomycetales (5.0%). The remaining classified bacteria (13%) belonged to the genus *Bacillus* (Firmicutes). It is important to bear in mind that the bacterial isolation procedures used in the study were, in fact, directed to favor Actinobacteria, guided by the phenotypic characteristics of the colonies, therefore, the strategy applied herein has been demonstrated to be effective.

There are many studies describing the diversity of culturable Actinobacteria from marine sediments [22,23,24], as well as those recovered from marine invertebrates, including ascidians and sponges [14,25]. Indeed, all these previous studies pointed out the genera found in Rocas Atoll among the most common ones, especially in studies guided by cytotoxicity. The genus *Streptomyces* and the obligate marine genus *Salinispora* were greatly recurrent in the Rocas Atoll samples. The genus *Streptomyces* was recovered from all sample categories and was also the most diverse in terms of numbers of distinct lineages (Figure 2). On the other hand, out of the eight *Salinispora* strains, six were recovered from ascidians, while the other two were associated with sponges. Interestingly, these strains were more related to each other than with the three known species of the genus. The presence of *Salinispora* was recently described in Brazilian waters along with the associated chemical diversity expected for this genus, including the production of highly cytotoxic compounds such as saliniketals and rifamycins, within strains isolated from sediments from Saint Peter and Saint Paul Archipelago [26]. It is worth mentioning that *Streptomyces* bacteria are indigenous to most environments, and they have been recognized as one of the most skilled genera in terms of secondary metabolism, producing numerous clinically used drugs [27,28].

### 2.2. Chemical Diversity Produced by Bacteria from Rocas Atoll 

#### 2.2.1. Molecular Networking

The microbial diversity found in Rocas Atoll may be connected to an equivalent diversity of chemical structures and the dereplication process of crude extracts can assist in addressing this matter in an organized fashion. Thus, we applied herein molecular networking [29] combined with detailed analyses of the gas phase decomposition reactions observed in MS/MS spectra [30]. Also, the emergence and improvement of computational tools, along with public reference libraries, have improved the compound annotation procedure [31]. Therefore, the crude extracts produced by the recovered bacteria were analyzed by HPLC-MS/MS in order to create a molecular network to compare samples and to investigate the chemical diversity by dereplication of known compounds. In this step, all the crude extracts were included in the analyses, both cytotoxic and noncytotoxic. This approach was employed intending a prompt identification of molecular families present exclusively in cytotoxic extracts, focusing on metabolites that may be the responsible for the biological activity. Metabolites that also occur in noncytotoxic extracts might have decreased levels of cytotoxicity or have even been present at low concentrations in the crude extracts. The complete molecular network obtained for the bacterial extracts from Rocas Atoll is shown in Appendix A (after solvent blank removal), in which each node/sphere represents a parent ion of an MS/MS spectrum. The color code refers to bacteria identification according to the legend. All identified clusters of molecular family are amplified in Appendix A. To better visualize the data, Figure 3 highlights some clusters of molecular families that have at least one node annotated by the GNPS library [32]. Due to some false-positive results that automated spectral matches can generate, every spectrum from the data set that matched some spectrum from the GNPS library was manually checked, comparing with spectra from published papers.

The approach employed here allowed the annotation of many different chemical classes of compounds, such as diketopiperazines, lipopeptides (including surfactins and esperin), staurosporins, surugamides, sphinganines, erythromycins, TAN antibiotics, rifamycins, and the metal complexing agent (siderophore) desferrioxamine. These results disclosed the bacteria from Rocas Atoll as a rich source of compounds with biotechnological and pharmacological interest.

Diketopiperazines (DKPs) (Appendix A), which were detected in all crude extracts, are low-molecular-weight compounds known to display a large range of biological activities, comprising immunosuppression, antibacterial, antifungal, nematicidal, insecticidal, and cytotoxic against many different cancer cell lines [33,34], including a multidrug-resistant colon carcinoma, whose phenotype a DKP derivative was described as reverting [35]. Furthermore, DKPs have been reported to be involved in chemical signaling/communication between plants and plant-pathogenic bacteria [36], in which the production of these bacterial metabolites generates a range of functional responses in plants.

Although the evaluation of antimicrobial activity was not a goal of the present study, many chemical classes identified in this work have been described in the literature with this biological property. For instance, rifamycin antibiotics, like rifampin and rifapentine, are antimycobacterial substances produced by actinobacteria, and have been specifically used in the effective treatment of tuberculosis [37]. Here, rifamycin S, hydroxyrifamycin S, and dihydroxyrifamycin S, along with three unknown rifamycin analogs (*m/z* 662, 732, and 761) (Figure 3 and detailed in Appendix A) were produced only by *Salinispora* strains. On the other hand, lipopeptides, such as surfactins, are potent surfactant agents typically produced and mainly detected herein in extracts obtained from members of *Bacillus* genus. This chemical class presents a broad activity spectrum such as antibiotic agents due to the ability to penetrate cell membranes of Gram-positive and Gram-negative bacteria, as well as fungi [38]. In this study, lipopeptides (Figure 3 and detailed in Appendix A) were also detected in extracts from *Streptomyces*, *Salinispora*, and *Nocardiopsis* strains in lower amounts. Interestingly, in the molecular cluster of surfactins, there were some ions (*m/z* 980, 1024, 1032, 1042, 1052, and 1056) referring to molecular masses not described for the surfactin class, which shows that bacteria from Rocas Atoll might produce unknown surfactin analogs. Erythromycin is a macrolide widely employed as antibiotic agent to treat a number of bacterial infections [39]. It is usually produced by actinobacteria (e.g., *Streptomyces* and *Saccharopolyspora*). In this work, this chemical class (erythromycin A and 15-(2-Propynyl)erythromycin A) (Figure 3 and detailed in Appendix A) was detected mostly among noncytotoxic samples (BRB-407 and BRB-504), however, a possible unknown analog (*m/z* 716) was found in one extract produced by *Salinispora* (BRB-415). The potent class of antibiotic TAN was also detected in our samples (Figure 3 and detailed in Appendix A), produced predominantly by *Nocardiopsis* and some noncytotoxic extracts, but derivatives were also produced by members of Firmicutes, *Streptomyces*, *Salinispora*, and nonidentified strains. Members of this dipeptide antibiotic family have presented activity against Gram-positive and Gram-negative bacteria, especially against drug-resistant strains [40].

Still regarding peptides, the detection of surugamides within the Rocas Atoll bacterial extracts should be highlighted (Figure 3 and detailed in Appendix A). These cyclic octapeptides produced by marine *Streptomyces* were only recently described and are still poorly studied. In the generated molecular network, a cluster containing seven members of surugamides were detected mainly in noncytotoxic extracts and extracts produced by *Streptomyces* strains, but also by one *Nocardiopsis* strain (BRB-352). These compounds are described as inhibitors of cathepsin B, a cysteine peptidase overexpressed in many pathological events, such as inflammation and cancer. Cathepsin B inhibitors are promising compounds to be applied in cancer treatments [41]. To date, there are only five surugamides described in the literature. Our results indicate the presence of unknown surugamide analogs in the Rocas Atoll samples, thus suggesting that these bacteria are an interesting source of new chemicals to be further investigated.

Compounds previously described with cytotoxic activity, staurosporines (Figure 3 and detailed in Appendix A) and saliniketals (as single node in Appendix A), were also identified in this work. The putative staurosporine and derivatives have been extensively studied for years regarding their anticancer properties [42,43]. Staurosporine and derivatives were previously detected by our research group and isolated from the ascidian *Eudistoma vannamei* collected on the west coast of Ceará state, Brazil [44,45]. This chemical class was detected in extracts from *Salinispora* and *Streptomyces* strains, but also in noncytotoxic strains. Moreover, saliniketals A and B, two unusual bicyclic polyketides, produced by the marine actinobacteria *Salinispora arenicola*, have shown inhibition of ornithine decarboxylase, a target for cancer chemoprevention therapies [46]. Here, saliniketal A was detected in most extracts produced by *Salinispora* strains. Although this chemical class was not present as a cluster in the molecular network, this compound was detected and annotated by the GNPS library.

#### 2.2.2. Chemometric Analysis

A total of 3985 ions were detected, of which 1642 were common to both active and nonactive extracts, and 1475 were exclusive to active ones (Appendix A). In general, there were more ions in the cytotoxic samples than in the noncytotoxic ones, and while the cytotoxic strains produced an average of 471 ± 40 ions/strain, noncytotoxic ones produced 224 ± 24 putative metabolites, representing a rate of approximately 2:1 (Appendix A). The Shannon–Wiener index (H′) (Appendix A) shows that the cytotoxic extracts (H′ = 5.7 ± 0.1) displayed a more diverse metabolomic content than noncytotoxic extracts (H′ = 4.9 ± 0.1).

In addition to what was observed through analysis of the molecular network, a relationship between the metabolomic profile and taxonomy became evident. The production of molecular families, such as lipopeptides, can be clearly associated with *Bacillus*. These findings are somewhat expected, since differences in the genomes lead to the expression of different enzymes and biosynthetic pathways, which in turn generate distinct chemical structures. In the heatmap (Figure 4), there are clear associations between the chemical groups with a particular genus, such as the antibiotic TAN and another unknown molecular family that were detected in all *Nocardiopsis* strains; as well as rifamycin, staurosporine, and another unknown molecular family that were detected in most *Salinispora* strains.

### 2.3. Molecular Networking vs. Cytotoxicity

The molecular network analysis was further applied in order to highlight clusters of ions more likely to be accountable for the biological activity of the extracts, since those present in both cytotoxic and noncytotoxic samples would hardly be bioactive metabolites. Such a strategy allowed efforts to be directed toward some few clusters, decreasing the number of possibilities and pinpointing the most interesting groups of compounds to be investigated. Among the identified molecular families, rifamycins, lipopeptides, and staurosporines should be emphasized. Besides those, one cluster stood out by grouping compounds present exclusively in samples considered cytotoxic (circled in light blue in Appendix A). This cluster comprised parent masses in the range of 800 to 1400 *m/z* (Appendix A), produced specifically by BRB-298 and BRB-302 strains, both identified as *Streptomyces* sp. The MS/MS data showed a fragmentation pattern of glycosylated structures. These data presented no matches in the GNPS library or even other public libraries that were searched, such as “Dictionary of Natural Products” and “SciFinder”, indicating a high probability of being novel compounds, but future investigations should be undertaken to confirm this possibility. Therefore, the BRB-302 strain was selected to be cultured in large scale and fractionated, in order to investigate the cytotoxicity of the fractions and to check if the compounds highlighted in the molecular network are indeed the active compounds in the crude extract.

### 2.4. Glycosylated Compounds Produced by BRB-302

The approach presented in this work allowed the selection of one crude extract among 76, and furthermore, it allowed the active compounds in the crude extract to be pointed out without the need for isolation. The crude extract was fractionated, and the cytotoxicity of the fractions were evaluated. Fractions 28–32 presented the highest cytotoxicity and the LC-MS/MS showed that these fractions contained those glycosylated ions highlighted before in the molecular network. Therefore, in order to identify at least the chemical class of those unknown compounds produced by the BRB-302 strain without isolating them, we decided to continue exploring our data employing other metabolomic tools available at GNPS. Molecular network is a tool at GNPS for organization and visualization of untargeted data, and it also allows an automatic spectral library search, which has contributed to the annotation of known compounds in metabolomics [32]. However, GNPS also comprises several other computational tools, including some in silico, that can improve the annotation, especially considering known compounds that do not present available MS/MS spectra, or even unknown compounds. In general, predictive tools have been successfully applied to the annotation of several challenging structures from different chemical classes [47,48]. In this context, DEREPLICATOR+ is an algorithm able to construct theoretical MS/MS spectra of compounds from natural products structure databases, including peptides, polyketides, flavonoids, terpenes, and alkaloids, among other classes of natural products, through cleaving the molecular bindings of such compounds [49]. 

Therefore, we analyzed the MS/MS data of the extract produced by the *Streptomyces* sp. BRB-302 on DEREPLICATOR+. The tool indicated a structure for only one component in the entire cluster, the novonestmycin A (molecular mass 1,228.6605 Da) [50], which was designated for the ion *m/z* 1,229.6635 ([M + H]^+^, error 4.0 ppm). UV(MeOH) λ_max_ described for novonestmycins are 222, 260, and 290 nm, while the observed for the compound produced by *Streptomyces* sp. BRB-302 are 222, 261, and 290 nm (Figure 5). Moreover, novonestmycins are produced by *Streptomyces phytohabitans*, and both *Streptomyces* sp. BRB-298 and BRB-302 strains are on the same branch of *S. phytohabitans* in the phylogenetic tree (Figure 2). Furthermore, novonestmycins were reported as highly cytotoxic against tumor cell lines, as well as the extract obtained from *Streptomyces* sp. BRB-302 and the active fractions. Examining the fragmentation spectrum of *m/z* 1,229.6635, the neutral loss of 280 a.m.u. was observed, which refers to the loss of the phenolic glycoside moiety, followed by successive water losses, which is a pattern for polyketide macrolide chains [51,52]. Therefore, as our data presented different masses than those described in the literature for novonestmycins, we can assume that *Streptomyces* sp. BRB-302 and BRB-298 produce several novel novonestmycins derivatives, which will be the aims of forthcoming studies concerning isolation, structure characterization, and bioactivity assessments. Here, we focused on the chemical richness produced by the recovered bacteria from Rocas Atoll. Furthermore, metabolomics approaches allowed disclosing the possible chemical class of compounds responsible for the cytotoxicity observed in the extracts of *Streptomyces* sp. BRB-302 and BRB-298 strains, which is a motivating strategy to be employed in screening of natural crude extracts.

## 3. Materials and Methods 

### 3.1. Sample Collection

Different ascidian and sponge specimens and also sediment samples were manually collected at Rocas Atoll (03°51′00″ S and 33°49′00″ W). The samples were immediately frozen in liquid nitrogen for transportation and kept in the laboratory at −20 °C until they were processed. The project was developed under permission from the Instituto Chico Mendes de Conservação da Biodiversidade (SISBIO # 44435-1), as well as from the Conselho Gestor do Patrimônio Genético (SISGEN # 010170/2015-4).

### 3.2. Bacteria Isolation and Cultivation

Under sterile conditions, invertebrate specimens were sorted for epibionts and debris and then dipped in ethanol 70% for decontamination of the surface. Next, all samples (ascidians, sponges, and sediments) were diluted in sterile seawater (1:1 w/v), heated to 55 °C over 10 min and inoculated onto 90 mm Petri dishes filled with one of three types of medium: a nutrient-rich medium (A1); trace metals agar (TMA), a nutrient-poor medium with trace metals; and seawater agar (SWA), a minimum medium. All media were prepared with reconstituted seawater, and cycloheximide (0.1 mg/mL) was added to reduce fungal contamination. The use of these three different culture media was important to recover bacterial strains with different metabolic requirements, since the use of nutrient-rich media can underestimate the diversity of culturable Gram-positive bacteria due to the overgrowth of fast-growing strains, while the use of low-nutrient and minimum media generally improve isolation of diverse microorganisms [53].

Sediment-inoculated dishes were left between 12 and 90 days at 26–28 °C to allow for bacterial growth. Distinguishable colonies were selected based on their phenotypic characteristics, including color, brightness, shape, and texture, then transferred to fresh A1 (prepared in the laboratory with soluble starch, yeast extract, and peptone) agar dishes using a sterile toothpick, for isolation and purification. Pure strains were further grown in liquid media, supplemented with 25% glycerol, codified and distributed into cryovials for preservation at −80 °C. The isolated bacteria were coded by BRB followed by a three-digit number.

### 3.3. Crude Extract Production

Bacterial strains were grown in Erlenmeyer flasks (250 mL) containing 50 mL of A1 liquid medium. The cultures were maintained at 28 °C under 170 rpm between 3 and 25 days, depending on the proliferation rate of each strain under the offered conditions. The cultures were extracted with ethyl acetate (50 mL) over 2 h and the organic fraction was collected. Solvent was then removed under reduced pressure (rotary evaporator) to yield the respective crude extracts.

### 3.4. Cultivation and Fractionation of BRB-302 Strain

Cryovials of strain BRB-302 were inoculated into two Erlenmeyers (2 L) containing 250 mL of A1 medium and cultivated for seven days at 180 rpm and 28 °C. Cultivated broths were extracted twice with 200 mL of ethyl acetate, kept at 180 rpm for 1 h. Crude extracts were combined (55 mg), diluted to a concentration of 50 mg/mL, and fractionated through analytical HPLC-PDA.

Chromatographic separation was carried out through a chromatograph Agilent 1200 containing a fraction collector and using a Zorbax^®^ Eclipse Plus C18 4.6 × 150 mm, 3.5 µm column at 45 °C. The method consisted of solvents A (0.1% acetic acid in H_2_O) and B (MeOH), starting at 5% up to 100% of B in 30 min followed by a hold of 100% of B for 5 min. Fractions were collected at 0.75 min intervals over 30 min from the beginning of the chromatographic run, and one at the end of the fractionation (100% MeOH), for a total of 41 fractions collected.

### 3.5. Cytotoxic Assay

The crude extracts and fractions were dissolved in DMSO and evaluated regarding their cytotoxicity at two different concentrations, 5 and 50 µg/mL, against a colon carcinoma cell line (HCT-116 ATCC CCL-247), using the MTT assay [54]. Briefly, cells cultured in RPMI media supplemented with 10% SBF were plated into 96-well plates (10,000 cells per well), left to adhere for 24 h, and exposed to test samples, in duplicates, over 72 h. Doxorubicin and DMSO were used as positive and negative controls, respectively. After this period, wasted media were substituted with fresh media containing MTT at 0.5 mg/mL. After 3 h, media were removed, the reduced MTT (formazan) was dissolved in DMSO, and the absorbance was measured at 570 nm. Single-well data were transformed to percentage of growth inhibition after normalization with positive (100% of inhibition) and negative (0% of inhibition) controls. Extracts were considered cytotoxic when they inhibited over 75% cell growth at 50 µg/mL.

### 3.6. DNA Extraction, 16S rRNA Amplification, and Sequencing

Strains producing cytotoxic crude extracts were selected for taxonomic identification. Firstly, a phenotypic analysis, including the morphology of bacterial colony, and the Gram test were performed. Then, the genomic DNA from the bacterial culture was isolated, following the Wizard^®^ Genomic DNA Purification Kit, with some modifications, such as the use of Proteinase K, a higher concentration of RNase, and a greater incubation period during the extraction step [53,55]. Partial amplification of the 16S rRNA gene was performed with Promega’s GoTaq Green Master Mix, and universal primers for eubacteria 27F (5′-AGAGTTTGATCCTGGCTCAG-3′) and 1492R (5′-TACGGCTACCTTGTTACGACTT-3′). PCR products were purified with a purification kit (QIAGEN Inc., Valencia, CA, USA) and sequenced using the ABI 3730 DNA Analyzer, a 48-capillary DNA analysis system with Life Technologies technology (Applied Biosystems, Foster City, CA, USA). Sequencing reactions were performed using the BigDye^®^ Terminator v3.1 Cycle Sequencing Kit (code 4337456, Thermo fisher Scientific, Carlsbad, CA, USA) and runs carried out in capillaries (36 cm) using the POP7 polymer. Sequences were analyzed using ChromasPro software (version 2.0, TECHNELSIUM DNA Sequencing Software, Australia, http://technelysium.com.au/wp/chromaspro/), compared to the sequences present in the Ezbiocloud database for type species and also at NCBI (http://www.ncbi.nlm.nih.gov/) using Basic Local Alignment Search Tool (BLAST).

### 3.7. Phylogenetic Analysis 

The forward and reverse sequences from the isolated bacteria strains were aligned and the consensus sequences were obtained using Geneious 7 software (Biomatters Ltd, New Zealand). All the sequences reported in this study were deposited in GenBank under accession numbers MK720152 through MK720174 (Appendix A) (http://www.ncbi.nlm.nih.gov/genbank). Sequences were compared to the EzBioCloud database (http://www.ezbiocloud.net), and most similar sequences from type species were included in the analysis. The full set of sequences was then aligned with MAFFT version 7 using the L-INS-i strategy [56]. For the phylogenetic reconstruction, we used the GTR+G nucleotide substitution model. The phylogenetic relationships were inferred through a maximum likelihood approach using RaxML version 8 [57] with 1000 bootstrap iterations. The resulting tree was exported and edited using the iTOL v4 online tool [58,59].

### 3.8. Metabolomic Fingerprint by HPLC-MS/MS

Crude extracts and fractions were diluted in methanol at 1.0 mg/mL and analyzed by HPLC-MS/MS for metabolomics investigation. The analyses were developed on an HPLC system (Shimadzu) coupled to a micrOTOF QII QqTOF mass spectrometer (Bruker Daltonics, Billerica, MA, USA) fitted with an electrospray ionization source, operating in positive ionization mode. The chromatographic separation occurred on a Supelco Ascentis Express C18 (5 μm, 150 × 3.0 mm) column, using water (phase A) and acetonitrile (phase B) as the mobile phase, both containing 0.1% formic acid. A gradient was employed from 5% to 100% phase B over 25 min followed by 100% phase B for 6 min, with a flow rate of 0.7 mL/min. The column temperature was set to 40 °C. Ion source: ESI, endplate: 4500 V, capillary voltage: 3500 V, dry temperature: 220 °C, dry gas: 9.0 L/min, *m/z* range: 50–1500, gas pressure: 40 psi. The injection volume was 15 µL. An untargeted method was used in the mass spectrometer, in which the analyzer selected the higher intensity ions to fragment. A ramp from 20 to 75 eV as collision energy was employed.

### 3.9. Molecular Networking 

HPLC-MS/MS data were converted to mzXML file format by using MSConvert, and thereafter, uploaded to the Global Natural Products Social (GNPS) molecular networking server (gnps.ucsd.edu) [32]. On the GNPS platform, MS/MS spectra were combined with MSCluster algorithm considering cosine similarity values (higher than 0.95) to create consensus MS/MS spectra. For spectral networks, parent mass and fragment ion tolerance of 0.08 Da and 0.1 Da were considered, respectively. For creating edges, a cosine score over 0.65, more than four matched peaks, and two nodes at least in the top 10 cosine scores (K parameter) were fitted. The spectra were also searched against GNPS’s spectral libraries, considering scores above 0.65 and at least four matched peaks. Data are publicly available at http://gnps.ucsd.edu under accession number MSV000083601. The generated molecular network was imported and visualized as nodes and edges into Cytoscape version 3.4.0 [60]. Nodes represent parent masses and edge thickness corresponds to cosine score between two nodes.

Data used in the molecular network were recovered as a .csv table and analyzed to obtain the number and abundance of precursor ions among cytotoxic and noncytotoxic extracts. The Shannon–Wiener diversity index was calculated based on the molecular abundance of the precursor ions using R software version 3.6.1 with the ’vegan’ package version 2.5-6. To verify the significance of retrieved value, data from cytotoxic and noncytotoxic extracts were compared with Student’s *t* test with a level of significance of 5% using GraphPad Prism.

The ions/nodes referring to annotated chemical classes were organized in a table containing the number of analogs produced by individual strains. These data were used to construct the heatmap using R software version 3.6.1 (The R Foundation, Austria). The bacterial order was kept as generated from the phylogenetic tree (described under topic 3.6), while the chemical classes were clustered based on Ward’s method employing Euclidian distance.

## 4. Conclusions

Rocas Atoll houses a marine bacteria community skilled in producing metabolites with interesting pharmacological properties. The approach employed herein was efficient in allowing the detection and annotation of several chemical classes in extracts produced by the recovered bacteria. Particularly, such a breakthrough was made possible due to the versatility of mass spectrometry, which generated a large amount of data on the crude extracts, including important structural information on organic compounds therein, which accelerated their annotation in complex samples. Molecular network approaches employed here successfully indicated which are the metabolites in the entire dataset responsible for the activity observed for the crude extracts, with no need of isolation steps. Additionally, the tools available at GNPS platform were very useful; besides molecular network, spectral library search and DEREPLICATOR+ accelerated and allowed the annotation of the compounds of interest in this work. Overall, we may infer that the equatorial Brazilian islands host novel, peculiar, and diversified functional natural products and, moreover, the evidence raised herein support further and favorable claims for the protection of these regions against anthropic impacts to prevent ecosystem degradation and the loss of such valuable resources.

## Figures and Tables

**Figure 1 marinedrugs-17-00671-f001:**
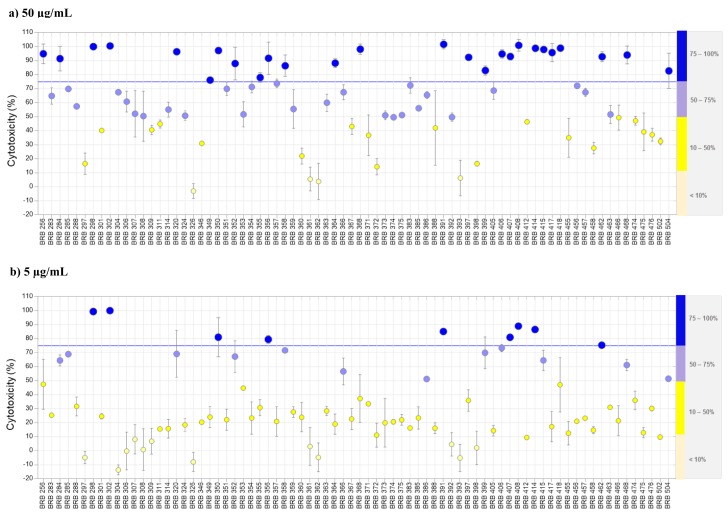
Cellular viability. Cytotoxicity of crude extracts produced by the recovered bacteria against HCT-116 carcinoma cell lines at 50 µg/mL (**a**) and 5 µg/mL (**b**), evaluated by the MTT assay after 72 h exposure. The extracts that presented cytotoxicity above 75% at 50 µg/mL were considered active. Among 76 extracts tested, (**a**) 26 were cytotoxic at 50 µg/mL, and (**b**) nine were also cytotoxic at 5 µg/mL.

**Figure 2 marinedrugs-17-00671-f002:**
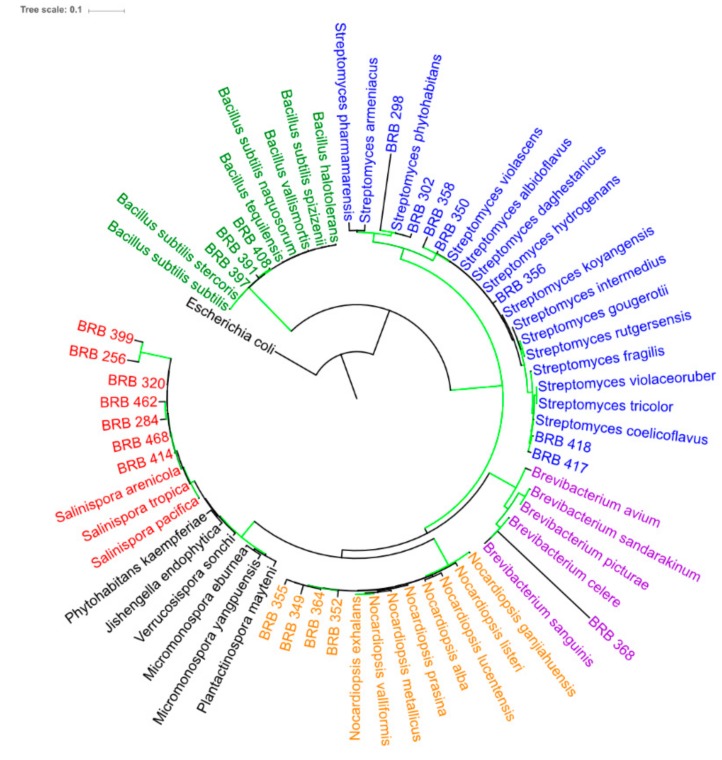
Phylogenetic relationships among recovered strains and most similar type species of bacteria obtained through Maximum Likelihood (RaxML) with 1000 bootstrap replications. Green lines indicate bootstrap support values > 70% for the branch. Terminals are colored according to the bacteria genera and follow the same pattern throughout this article. Accession numbers are shown on Appendix A. The strains recovered from Rocas Atoll are coded by BRB followed by a three-digit number.

**Figure 3 marinedrugs-17-00671-f003:**
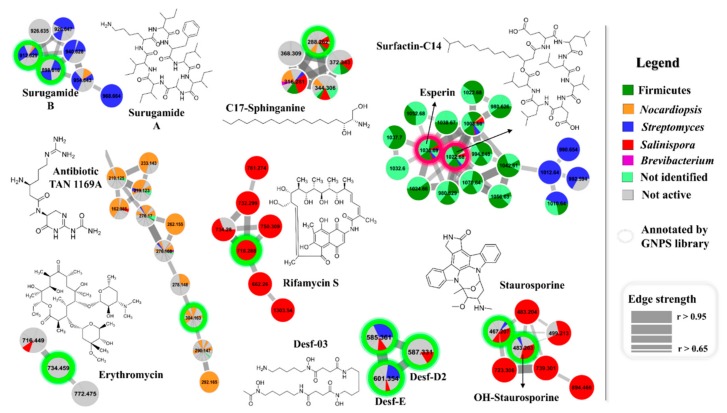
Molecular network of extracts produced by bacteria recovered from Rocas Atoll, considering the positive ionization mode (ESI+) data after removal of solvent blank. Nodes represent parent masses, and their colors are according to the legend, where the noncytotoxic extracts (<75%) are in gray. Extracts that showed inhibition of cell growth above 75% are divided by taxonomic group in Actinobacteria (*Streptomyces*, *Salinispora*, *Nocardiopsis*, and *Brevibacterium*) and Firmicutes (*Bacillus*). Extracts considered cytotoxic produced by bacteria nonidentified are depicted in light green. Only clusters containing at least two nodes are shown.

**Figure 4 marinedrugs-17-00671-f004:**
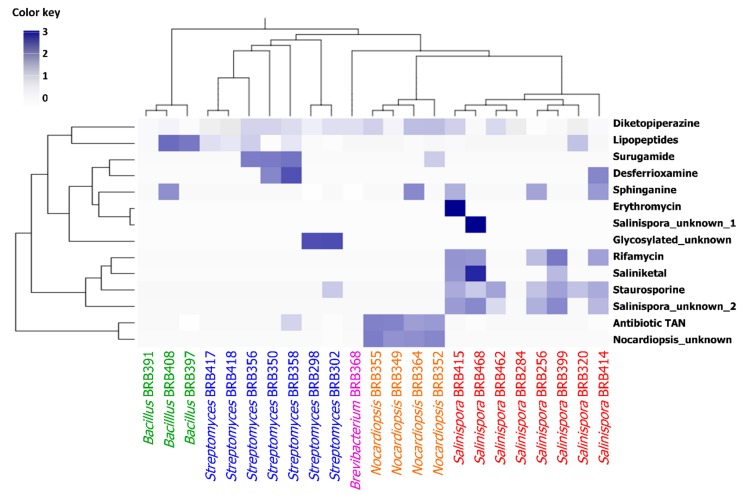
Heatmap showing the distribution of metabolites classes vs. bacterial phylogenetic tree. Only the bacteria that produced cytotoxic samples were considered. The chemical classes used here were obtained from the spectral library annotations. The ions/nodes referring to identified chemical classes were used to construct the heatmap using software R. The color represents the number of analogs (normalized by logarithmic scale) each strain produced for the respective chemical classes. The clustering was based on Ward’s method employing Euclidian distance.

**Figure 5 marinedrugs-17-00671-f005:**
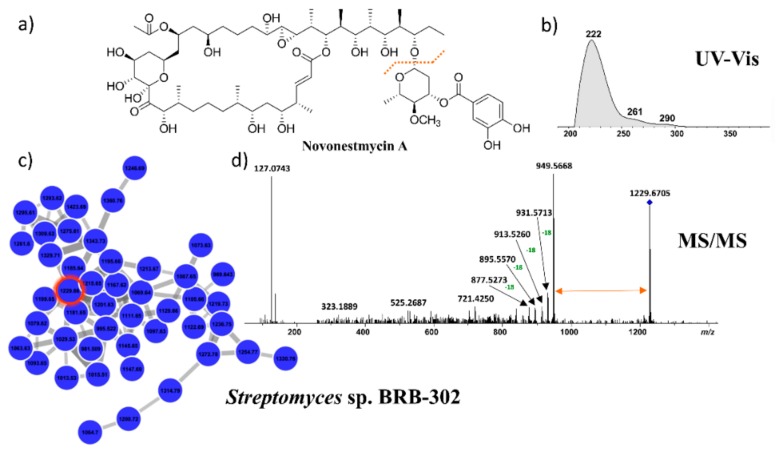
(**a**) Molecular structure of novonestmycin A; (**b**) UV spectrum referring to the novonestmycin derivative produced by the *Streptomyces* sp. BRB-302; (**c**) Cluster of an active molecular family produced exclusively by *Streptomyces* sp. BRB-298 and BRB-302 strains, in which the compound annotated by DEREPLICATOR+ is circled in red; (**d**) MS/MS spectrum of the ion *m/z* 1,229 annotated as novonestmycin derivative.

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
