# Peer review of "Marine Bacteria from Rocas Atoll as a Rich Source of Pharmacologically Active Compounds"

_marinedrugs, 2019, doi:10.3390/md17120671_

Round 1
Reviewer 1 Report
The manuscript presents phylogenetic and metabolomic analysis of cultures collected from an island. The authors test the cytotoxicity of the cultures using one assay and follow up by presenting a mass spectrometry-based assessment of the organic compounds. I think that the combination of cultures, toxicity assays, and mass spectrometry is useful.
My biggest concern with the manuscript is the authors’ over-reliance on the results from GNPS as ‘identifying’ unknown compounds. GNPS provides a valuable framework to explore mass spectrometry data; however, the results are not sufficient to identify unknown compounds. Nowhere in this manuscript do the authors present any indication that there are limitations to the molecular networking approach. Instead the data are presented as certain identifications which I feel overstates the output from GNPS. At best, the compounds discussed are ‘putative annotations’ within the framework established by Sumner et al. (2007, "Proposed minimum reporting standards for chemical analysis." Metabolomics 3(3): 211-221.) The authors should read the guidelines established by Sumner and use that to discuss the putative annotations they find within their dataset.
The manuscript would also benefit from a more thorough review for grammar and language use.
Minor comments:
In the introduction, the inconsistent use of the abbreviation for Rocas Atoll (RA) is confusing. I would not abbreviate the word as it is such a central portion of the manuscript.
Line 34-35: ‘molecular networking’
Line 36: with this phrasing, it is not clear what GNPS is an abbreviation for in this sentence.
Line 100: I am not familiar with this cytotoxicity assay, so the phrasing used here is a little confusing. From Figure 1, the degree of cytotoxicity is given as a percentage. Therefore, presenting the number of strains that exceeded the 75% threshold as a percentage is confusing because percentage is used in two different ways in the same paragraph.
Line 126: Are the strains that begin with BRB the ones from the present project? This would be helpful to note in the figure legend.
Line 171: ‘Besides, the higher the number of spectra, the greater the chance of annotation of known compounds.’ This is not a complete sentence.
Line 181: what did the manual checking entail, and what metrics were used to determine if a putative annotation was or was not valid?
Line 201: I do not see any light blue in this figure, so I am not clear what the authors are referring to.
Line 278 ‘ one large cluster stood out’ …this is vague. I am not clear why the cluster stood out, or why the authors chose to focus on this cluster.
Line 279: I am confused because figure S2 has at least four clusters in light blue, but the text here only describes one cluster circled in light blue. Also, how does figure S13 indicate the compounds are only produced by Streptomyces? That figure is a TIC plot and growth inhibition, not a figure indicating which bacterial group is producing the compounds of interest.
Line 282-284: ‘Moreover, an in-source fragmentation was observed, generating many parent masses for the same compound, which is characteristic of compounds with many glycosyl moieties.’ Where is this shown, and what is the evidence that this is characteristic of glycosyl moieties?
Line 313: What is the resolution of the QTOF used in this project? I suspect it is quite a bit larger than the 3.4 ppm error provided here for this potential annotation. Given this, I am concerned that the authors are over-stating their annotation for this compound.
Figure 4: for the ‘relative abundance’, is this the number of metabolites normalized for each species (by columns) or for each metabolite (by rows). This is not clear from the figure legend or the accompanying text.
Figure S2 – the legend provides the species from which the extract originated. However, the nodes are so small that it appears that none of the nodes originate from multiple species. In Figure 3 this is clearer as there are small pie charts, but these pie charts are lost in Figure S2 because they are so small.
Figures S3 – S11: what are the green or pink circles around some of the nodes?
Figure S12: the labels on top of the Venn diagram are truncated. The yellow in subplots (B) and (C) is difficult to see, a darker color would be easier. Finally, the statement that the ‘.csv table was considered’ is too vague and I am not certain what analysis was done. How was the table used? What are the statistics? Without information about what is in the table, as a reader I have no idea how the table was ‘considered’.
Author Response
Comments and Suggestions for Authors
The manuscript presents phylogenetic and metabolomic analysis of cultures collected from an island. The authors test the cytotoxicity of the cultures using one assay and follow up by presenting a mass spectrometry-based assessment of the organic compounds. I think that the combination of cultures, toxicity assays, and mass spectrometry is useful.
We thank the reviewer for the considerations made to improve our manuscript, and each point raised has been addressed in the following section.
My biggest concern with the manuscript is the authors’ over-reliance on the results from GNPS as ‘identifying’ unknown compounds. GNPS provides a valuable framework to explore mass spectrometry data; however, the results are not sufficient to identify unknown compounds. Nowhere in this manuscript do the authors present any indication that there are limitations to the molecular networking approach. Instead the data are presented as certain identifications which I feel overstates the output from GNPS. At best, the compounds discussed are ‘putative annotations’ within the framework established by Sumner et al. (2007, "Proposed minimum reporting standards for chemical analysis." Metabolomics 3(3): 211-221.) The authors should read the guidelines established by Sumner and use that to discuss the putative annotations they find within their dataset.
After reading the reviewer's comments we completely agree that the most appropriate term would be annotation rather than identification. In addition, we agree that the first paragraph of the results (section 2.2.1. Molecular networking evaluation) would be more appropriate for an introduction to an article on the technique development. In this sense we removed the paragraph and only include two sentences in the following paragraph to explain which strategy was used.
The manuscript would also benefit from a more thorough review for grammar and language use.
English was revised by a native speaker.
Minor comments:
In the introduction, the inconsistent use of the abbreviation for Rocas Atoll (RA) is confusing. I would not abbreviate the word as it is such a central portion of the manuscript.
We changed RA for Rocas Atoll throughout the entire manuscript.
Line 34-35: ‘molecular networking’
Done.
Line 36: with this phrasing, it is not clear what GNPS is an abbreviation for in this sentence.
It was addressed at lines 31-32: “The search in the spectral library in GNPS (Global Natural Products Social Molecular Networking)…”
Line 100: I am not familiar with this cytotoxicity assay, so the phrasing used here is a little confusing. From Figure 1, the degree of cytotoxicity is given as a percentage. Therefore, presenting the number of strains that exceeded the 75% threshold as a percentage is confusing because percentage is used in two different ways in the same paragraph.
We changed the information to the number of strains producing active extracts instead of percentage, since cytotoxicity is generally presented as a percentage of inhibition of cell growth. The assay employed here to evaluate cytotoxicity of the crude extracts is a very well-known and widely used assay called MTT, and the detailed explanation for it is under topic 3.5 in the ‘Material and Methods’ section. Briefly, MTT is a colorimetric assay based on the conversion of a water-soluble tetrazolium salt to the formazan purple precipitate by the viable cells. The intensity of the color, thus, is proportional to the number of viable cells, and the cytotoxicity is calculated based on the reduction of the absorbance readings compared to that observed for negative controls containing cells exposed only to the vehicle (DMSO) at the same concentration used to dilute the extracts.
Mosmann, T., Rapid colorimetric assay for cellular growth and survival: application to proliferation and cytotoxicity assays. Journal of immunological methods 1983, 65, (1-2), 55-63.
Line 126: Are the strains that begin with BRB the ones from the present project? This would be helpful to note in the figure legend.
To clarify it, the following sentences were added to: 1) the figure legend: “The strains recovered from Rocas Atoll are coded by BRB followed by a three-digit number”; and 2) in the ‘Material and Methods’ section: “The isolated bacteria were coded by BRB followed by a three-digit number”.
Line 171: ‘Besides, the higher the number of spectra, the greater the chance of annotation of known compounds.’ This is not a complete sentence.
To avoid confusion, we removed this sentence.
Line 181: what did the manual checking entail, and what metrics were used to determine if a putative annotation was or was not valid?
Actually, the library search realized through molecular network employs a method called “Variable dereplication”. This method calculates a vector for each ion considering the fragment ions values and their intensity. It then calculates the cosine score between vectors. By default, it considers for annotation those spectra that present a cosine score above 0.7; spectra with a cosine score above 0.9 are considered as being from the same compound. However, as a computational algorithm, it can be wrong sometimes, thus, we manually checked all the matches, comparing with spectra from published papers, then discarded those annotations that did not make sense. The dataset and annotations are publicly available at http://gnps.ucsd.edu under accession number MSV000083601, thus, readily available if one wishes to conduct a personal examination.
Line 201: I do not see any light blue in this figure, so I am not clear what the authors are referring to.
It’s light green. It was corrected in the figure legend.
Line 278 ‘ one large cluster stood out’ …this is vague. I am not clear why the cluster stood out, or why the authors chose to focus on this cluster.
The sentence was clarified: “Besides those, one cluster stood out joining compounds present exclusively in samples considered cytotoxic (circled in light blue in Supplementary Figure S2). Such cluster comprised parent masses in the range of 800 to 1400 m/z, approximately (Supplementary Figure S13), and grouped compounds produced specifically by strains BRB-298 and BRB-302, both identified as Streptomyces sp.”
Line 279: I am confused because figure S2 has at least four clusters in light blue, but the text here only describes one cluster circled in light blue. Also, how does figure S13 indicate the compounds are only produced by Streptomyces? That figure is a TIC plot and growth inhibition, not a figure indicating which bacterial group is producing the compounds of interest.
The Figure was changed and only one cluster is in light blue.
Line 282-284: ‘Moreover, an in-source fragmentation was observed, generating many parent masses for the same compound, which is characteristic of compounds with many glycosyl moieties.’ Where is this shown, and what is the evidence that this is characteristic of glycosyl moieties?
We agree with the reviewer concern about this sentence. Actually, this is just a speculation and we cannot assume that. In this work, we observed that some parent ions from this cluster (at the same retention time) presented m/z as if they lost one sugar moiety. In-source fragmentation is very common for O-glycosylated compounds (Pilon et al., 2019), and we have observed this a lot in our laboratory. However, the reviewer concern is right and we know that both compounds might be present in the crude extract. Therefore, we decided to remove this sentence from the manuscript. PILON, Alan Cesar et al. Mass spectral similarity networking and gas-phase fragmentation reactions in the structural analysis of flavonoid glycol-conjugates. Analytical chemistry, v. 91, n. 16, p. 10413-10423, 2019.
Line 313: What is the resolution of the QTOF used in this project? I suspect it is quite a bit larger than the 3.4 ppm error provided here for this potential annotation. Given this, I am concerned that the authors are over-stating their annotation for this compound.
This is not the error of the equipment, which can be as high as 20 ppm. Actually, we checked the value and, instead of 3.4 ppm, it is indeed 4.0 ppm, and it was corrected in the manuscript. This value refers to the error of the molecular mass detected (m/z 1,229.6635 [M+H]+, which means 1,228.6556 Da) in relation to the molecular mass of the annotated compound (novonestimycin 1,228.6605 Da), provided by the following formula:
Figure 4: for the ‘relative abundance’, is this the number of metabolites normalized for each species (by columns) or for each metabolite (by rows). This is not clear from the figure legend or the accompanying text.
In this figure we considered the number of analogs (nodes from the molecular network) detected for each chemical class described (by rows). In other words, it shows the number of analogs of a chemical class the strain is able to produces.
Figure S2 – the legend provides the species from which the extract originated. However, the nodes are so small that it appears that none of the nodes originate from multiple species. In Figure 3 this is clearer as there are small pie charts, but these pie charts are lost in Figure S2 because they are so small.
The figure was changed and amplified. The entire molecular network presented too many nodes and it was hard to check all of them in one figure, that is why we amplified, in Figure 3, the clusters we discussed throughout the manuscript. Moreover, the data is available online at http://gnps.ucsd.edu under accession number MSV000083601 (it is described in M&Ms), then the reader can examine if interested.
Figures S3 – S11: what are the green or pink circles around some of the nodes?
The circle highlights the node that was annotated from the Spectral library. A sentence was included in the figure legends to clarify the color pattern.
Figure S12: the labels on top of the Venn diagram are truncated. The yellow in subplots (B) and (C) is difficult to see, a darker color would be easier. Finally, the statement that the ‘.csv table was considered’ is too vague and I am not certain what analysis was done. How was the table used? What are the statistics? Without information about what is in the table, as a reader I have no idea how the table was ‘considered’.
The labels on top of the Venn diagram were corrected. However, we decided to keep the color yellow in the subplots, once this codification is in accordance with all other figures in the manuscript. The .csv table contains the ions detected in the sample, and it has been described in M&Ms. Besides, in the same legend, we described that the Venn diagram and dispersion of the ions were constructed with the information of the ions detected in cytotoxic and non-cytotoxic samples. We improved the sentence in the legend of this figure: “For these analyses, it was used the .csv table obtained from the molecular network containing the features information.” We also added to material and methods section “To verify the significance of observed value, data from cytotoxic and non-cytotoxic extracts were compared with Student’s t test with a level of significance of 5% using GraphPad Prism.”
Reviewer 2 Report
The authors investigate in their article marine bacteria from the Rocas Atoll and investigate the cytotoxic potential with regards to human colon adenocarcinoma cell line. The manuscript is written in a clear and structured way, the research design is appropriate and the findings are clearly communicated. Some improvement (typos, grammar, e.g. l. 209, 212, 217, and at a few other places) of English language is necessary.
Nonetheless, a few points should be addressed prior to publication.
title: words such as "promising" should not be used in the title, but rather replaced with somethin gthat more accurately describes the contents of the study title: the authors claim to have found a biotechnological valuable resource however the findings are based on the cytotoxity against one specific cell line. this is not enough to make such a broad judgement (especially since metagenomics/metatranscriptomics analysis was not conducted). please rephrase the title l.47: in 2016 alone -> please rephrase: only in 2016 l.51: reference (ref) missing (these comments always refer to the sentence that is ending in this line) l.53: ref missing l.62: ref is missing - why do these conditions provide a rich source of marine natural products? rich in contrast to what? l.66: considerable biodiversity: what does this mean? please rephrase and be more precise l.68: numbers less or equal than twenty should be written in words (please check in the entire manuscript) l.77: the term holobionts should be explained l.90: ref missing (considerin the great ... produced by marine bacteria) l.100-104: clearer separation between methods and results needed l.129: showed -> are shown l.148: such great microbial diversity -> comparison to similar habitats is missing l.157-159: please remove this sentence, this does not bring anything new to the manuscript, besides the need of computational support is depending on the user l.161-165: isn't this already stated in [31-33]? if so, then this can be removed. l.166-181: please separate methods and results l.203: why was antimicrobial activity not investigated (the whole paragraph deals with the topic) figure 4: scale (with numbers) is missing, it is unclear what the colours represent. it should also be explained more clearer in the text what the heatmap is showing/why it was done. l.309: ref missing l.351: which characteristics? l.403: which version of BLAST was used? l.409: version of Ez-Taxon database is missing l.413: version of RaxML is missing l.440: why two different main versions of cytoscape? l.445: version of R and vegan missing conclusions: remove the word treasured from line 458, apart from that, you should more focus the conclusions on the pharmacological potential, not biotechnology in general. there is not enough data presented to get to that conclusions.
Author Response
The authors investigate in their article marine bacteria from the Rocas Atoll and investigate the cytotoxic potential with regards to human colon adenocarcinoma cell line. The manuscript is written in a clear and structured way, the research design is appropriate and the findings are clearly communicated. Some improvement (typos, grammar, e.g. l. 209, 212, 217, and at a few other places) of English language is necessary.
We thank the reviewer for the considerations made to improve our manuscript, and each point raised has been addressed in the following section.
Nonetheless, a few points should be addressed prior to publication.
title: words such as "promising" should not be used in the title, but rather replaced with something that more accurately describes the contents of the study title.
the authors claim to have found a biotechnological valuable resource however the findings are based on the cytotoxity against one specific cell line. this is not enough to make such a broad judgement (especially since metagenomics/metatranscriptomics analysis was not conducted). please rephrase the title
We agreed with this ponderation and modified the title according to the suggestions. The new proposed title is: “Marine bacteria from Rocas Atoll as a rich source of pharmacologically active compounds”. Although we only evaluated the cytotoxicity against HCT-116 colon cancer cells using the MTT assay, the metabolomic analysis showed that bacteria from Rocas Atoll are producing several chemical classes with biological properties previously reported in the literature, such as surfactins, described as surfactant, surugamine, staurosporine and saliniketal, with anticancer properties, and the antibiotics rifamycin, surfactin, erythromycin and TAN. Based on the results, we suggested that bacteria from Rocas Atoll are a great source of compounds with pharmacological properties.
l.47: in 2016 alone -> please rephrase: only in 2016
Done.
l.51: reference (ref) missing (these comments always refer to the sentence that is ending in this line)
Actually references 1 and 2 were also related to this affirmative, thus we repeated the numbers in the end of this sentence.
l.53: ref missing
We included a reference to this sentence: Patricia Miloslavich, Eduardo Klein, Juan M. Díaz, Cristián E. Hernández, Gregorio Bigatti, Lucia Campos, Felipe Artigas, Julio Castillo, Pablo E. Penchaszadeh, Paula E. Neill, Alvar Carranza, María V. Retana, Juan M. Díaz de Astarloa, Mirtha Lewis, Pablo Yorio, María L. Piriz, Diego Rodríguez, Yocie Yoneshigue-Valentin, Luiz Gamboa, Alberto Martín. Marine Biodiversity in the Atlantic and Pacific Coasts of South America: Knowledge and Gaps. PLoS One. 2011; 6(1): e14631.
l.62: ref is missing - why do these conditions provide a rich source of marine natural products? rich in contrast to what?
The idea behind this affirmative is that genetic diversity, conveyed through biodiversity, generates chemical diversity. As the environment hosts a diverse biota, one can expect an associated chemical diversity. We added references about the unique biodiversity hosted in this environment (refs 3,4 and a new one - Alvarez-Yela AC, Mosquera-Rendón J, Noreña-P A, Cristancho M and López-Alvarez D (2019) Microbial Diversity Exploration of Marine Hosts at Serrana Bank, a Coral Atoll of the Seaflower Biosphere Reserve. Front. Mar. Sci. 6:338.
l.66: considerable biodiversity: what does this mean? please rephrase and be more precise
We rewrote this sentence: “Located 260 Km off the coast, Rocas Atoll is a unique environment, hosting many endemic species [3,4].”
l.68: numbers less or equal than twenty should be written in words (please check in the entire manuscript)
OK, we corrected throughout the manuscript.
l.77: the term holobionts should be explained
We included an explanation to holobiont: “These associations of the invertebrates with their associated microbial communities are considered complex ecosystems, called holobionts, which comprise a high microbial diversity composed predominantly by proteobacteria, but also by several other groups of bacteria and archaea [14-18].”
l.90: ref missing (considering the great ... produced by marine bacteria)
We included a reference: Boris Andryukov, Valery Mikhailov and Nataly Besednova. The Biotechnological Potential of Secondary Metabolites from Marine Bacteria. J. Mar. Sci. Eng. 2019, 7(6), 176; https://doi.org/10.3390/jmse7060176
l.100-104: clearer separation between methods and results needed
Done.
l.129: showed -> are shown
Done.
l.148: such great microbial diversity -> comparison to similar habitats is missing
We removed this sentence and only include two sentences in the following paragraph to explain which strategy was used.
l.157-159: please remove this sentence, this does not bring anything new to the manuscript, besides the need of computational support is depending on the user
We removed this sentence and only include two sentences in the following paragraph to explain which strategy was used.
l.161-165: isn't this already stated in [31-33]? if so, then this can be removed.
We removed this sentence and only include two sentences in the following paragraph to explain which strategy was used.
l.166-181: please separate methods and results
As previously mentioned in the answers of reviewer #1 comments, we reorganized the initial paragraph of this section, and left only the necessary information to explain which strategy was used.
l.203: why was antimicrobial activity not investigated (the whole paragraph deals with the topic)
Our screening program is focused on the study of anticancer activity, however the metabolomic and dereplication analysis suggested the presence of compounds with antimicrobial activity. Therefore, the paragraph discussed that the presence of such compounds, previously reported in the literature for their antimicrobial activity, is an interesting finding of our study.
figure 4: scale (with numbers) is missing, it is unclear what the colours represent. it should also be explained more clearer in the text what the heatmap is showing/why it was done.
We added the number of the scale and we also modified the legend of the figure to make the information in the figure more evident.
l.309: ref missing
We added a reference here: 31.Wang et al., Sharing and community curation of mass spectrometry data with Global Natural Products Social Molecular Networking. Nature biotechnology 2016, 34, (8), 828-837.
l.351: which characteristics?
We clarified this point. New text: The use of these three different culture media was important to recover bacterial strains with different metabolic requirements, since the use of nutrient-rich media can underestimated the diversity of culturable Gram-positive bacteria due to the overgrowth of the fast growing strains, while the use of low nutrient and minimum media generally improve isolation of diverse microorganisms [54].
l.403: which version of BLAST was used?
Basic Local Alignment Search Tool (BLAST) is available at NCBI website (http://www.ncbi.nlm.nih.gov/). According to the website: “BLAST finds regions of similarity between biological sequences. The program compares nucleotide or protein sequences to sequence databases and calculates the statistical significance.” There is no version specification associated to this program.
l.409: version of Ez-Taxon database is missing
Actually the corrected name of the database is EzBioCloud. According to the database description: “EzBioCloud is ChunLab's public data and analytics portal focusing on taxonomy, ecology, genomics, metagenomics, and microbiome of Bacteria and Archaea. Our new cloud service includes bioinformatics tools and succeeds our previous databases, which include EzTaxon, EzTaxon-e, and EzGenome”. There is no version specification associated to this program.
l.413: version of RaxML is missing
Information included (version 8).
l.440: why two different main versions of cytoscape?
In the beginning of this project, molecular network from GNPS could not be opened on recent version of Cytoscape, therefore, we opened it in the old version and then, create nice figures in the recent version. However, it is not necessary anymore, thus, we removed the oldest version from the manuscript.
l.445: version of R and vegan
We have included this information: R software version 3.6.1 and vegan package version 2.5-6
Missing conclusions: remove the word treasured from line 458, apart from that, you should more focus the conclusions on the pharmacological potential, not biotechnology in general. there is not enough data presented to get to that conclusions.
We rewrote this section. The new text is “Rocas Atoll house a marine bacteria community skilled in producing metabolites with interesting pharmacological properties. The approach employed herein was efficient in allowing for the detection and annotation of many different chemical classes produced by the bacteria. Particularly, such breakthrough was made possible due to the versatility of mass spectrometry, which generated a large amount of data on the crude extracts, including important structural information on organic compounds therein, which accelerated their annotation in complex samples. Additionally, the GNPS platform, a prodigious novelty in Natural Products Research, fast-tracked dereplication of known and unknown compounds and, furthermore, highlighted chemical groups possibly responsible for the biological activity observed for the crude extracts. Therefore, we may infer that the equatorial Brazilian islands, overall, host novel, peculiar and diversified functional natural products and, moreover, the evidences raised herein are further and favorable claims for the protection of these regions against anthropic impacts to prevent ecosystem degradation and the loss of such valuable resources.”
Round 2
Reviewer 1 Report
The authors have introduced new errors in spelling and grammar in their revision. The new version also needs to be corrected for English language usage.
The title in the supplementary information no longer matches the title in the main portion of the manuscript.
Line 36: correct to: compounds
Line 287: correct to: furthermore
Line 446: ‘…identified chemical classes…’ Per my previous review, these compounds have not been identified, therefore using the term ‘identified’ in the methods section is misleading.
Author Response
We thank the reviewer for the careful reading of our work. We corrected all the points raised and also carefully revised the English of the revised version.
The title in the supplementary information no longer matches the title in the main portion of the manuscript.
Corrected
Line 36: correct to: compounds
Corrected
Line 287: correct to: furthermore
Corrected
Line 446: ‘…identified chemical classes…’ Per my previous review, these compounds have not been identified, therefore using the term ‘identified’ in the methods section is misleading.
We changed to annotated as suggested.